# Real-World Treatment Efficacy and Safety Profile of Sofosbuvir- and Velpatasvir-Based HCV Treatment in South Korea: Multicenter Prospective Study

**DOI:** 10.3390/v17070949

**Published:** 2025-07-04

**Authors:** Jae Hyun Yoon, Chang Hun Lee, Hoon Gil Jo, Ju-Yeon Cho, Jin Dong Kim, Jin Won Kim, Ga Ram You, Sung Bum Cho, Sung Kyu Choi

**Affiliations:** 1Department of Gastroenterology and Hepatology, Chonnam National University Hospital and Medical School, Gwangju 61469, Republic of Korea; kjw9304@gmail.com (J.W.K.); choisk@jnu.ac.kr (S.K.C.); 2Department of Gastroenterology and Hepatology, Jeonbuk National University Hospital and Medical School, Jeonju 54896, Republic of Korea; chleemd@jbnu.ac.kr; 3Department of Gastroenterology and Hepatology, Wonkwang University Hospital and Medical School, Iksan 54538, Republic of Korea; jojo420600@gmail.com; 4Department of Gastroenterology and Hepatology, Chosun University Hospital and Medical School, Gwangju 61469, Republic of Korea; intelheal@gmail.com; 5Department of Gastroenterology and Hepatology, Jeju Halla Hospital, Jeju 63127, Republic of Korea; motet@hallahospital.com; 6Department of Gastroenterology and Hepatology, Hwasun Chonnam National University Hospital and Medical School, Hwasun 58128, Republic of Korea; rugaram27@hanmail.net (G.R.Y.); portalvein@naver.com (S.B.C.)

**Keywords:** hepatitis C virus, sofosbuvir, velpatasvir, voxilaprevir, efficacy, safety profile

## Abstract

Background: The advent of direct-acting antivirals (DAAs) has marked a significant milestone in the therapeutic landscape of hepatitis C, greatly improving treatment efficacy. A therapeutic regimen encompassing sofosbuvir (SOF), velpatasvir (VEL), and voxilaprevir (VOX) has demonstrated strong efficacy across all genotypes of the hepatitis C virus (HCV) and has recently been incorporated into the Korean healthcare system. This study aimed to evaluate the real-world efficacy and safety of these antivirals in the South Korean population. Methods: This prospective, multicenter, observational study enrolled patients with chronic HCV treated with SOF/VEL-based regimens at six hospitals between November 2022 and January 2024. DAA-naïve patients received SOF/VEL ± ribavirin for 12 weeks. Patients who had failed prior DAA therapy received SOF/VEL/VOX for 12 weeks. The primary endpoint was a sustained virological response at 12 weeks post-treatment (SVR12). Results: Among 101 patients treated with SOF/VEL, the mean age was 64.71 years, and 40.9% were male. Genotypes 1b and 2 were identified in 40.6% and 59.4% of patients, respectively. Two patients had a history of interferon-based treatment. The mean baseline HCV RNA level was 3,088,097 IU/mL. Cirrhosis was observed in 26.7% of patients (21.8% compensated; 5.0% decompensated). Of the 101 patients, 12 were lost to follow-up. Among the 89 patients who completed follow-up, SVR12 was achieved in 100.0% (89/89), including 5 patients with decompensated cirrhosis. In the SOF/VEL/VOX group, 17 patients were treated. The mean age was 61.84 years, 29.4% were male, and four had compensated cirrhosis. One patient was lost to follow-up. SVR12 was achieved in 100.0% (16/16) of the patients who completed follow-up. No serious adverse events (≥grade 3) were reported in either group during the DAA treatment period. Conclusions: In this first prospective real-world study in South Korea, SOF/VEL-based regimens demonstrated excellent efficacy and safety, achieving 100% SVR12 in the per-protocol population, including patients with cirrhosis and prior treatment failure.

## 1. Introduction

Hepatitis C virus (HCV) infection remains a global public health challenge, contributing significantly to morbidity and mortality. As of 2024, over 50 million people worldwide are estimated to have chronic HCV infection, which, if left untreated, can lead to liver cirrhosis, hepatocellular carcinoma, and other serious liver-related complications [1].

Despite advancements in antiviral therapies, effective HCV management has historically been limited by the complexity and side effects of early interferon-based treatments. However, the introduction of direct-acting antivirals (DAAs) has revolutionized HCV management, offering cure rates exceeding 90%, even in patients with advanced liver disease or previous treatment failure [2,3,4].

Among the available DAA regimens, the combination of sofosbuvir (SOF) and velpatasvir (VEL), which target the NS5B polymerase and NS5A proteins, respectively, has shown strong pangenotypic efficacy [5,6] and is recommended as a first-line HCV treatment by the European Association for the Study of Liver (EASL) and the American Association for the Study of Liver Diseases (AASLD). The additional inclusion of voxilaprevir (VOX), an NS3/4A protease inhibitor, further enhances treatment outcomes in patients who have failed previous DAA regimens. These combinations provide potent antiviral activity and achieve a sustained virological response (SVR)—the surrogate marker for HCV cure—across a broad range of patient populations [2,7].

In South Korea, hepatitis B virus (HBV) is the primary cause of chronic liver disease, followed by HCV as a significant viral etiology, with genotypes 1b and 2 predominating [8,9,10]. Notably, over 90% of affected individuals are over the age of 40 [11]. Recently, SOF/VEL and SOF/VEL/VOX regimens have been incorporated into South Korea’s national healthcare system. However, real-world data on their efficacy and safety in the Korean population are still limited [12].

This study aims to address this gap by evaluating the real-world efficacy and safety of SOF/VEL and SOF/VEL/VOX in South Korea. Specifically, we assess SVR at 12 weeks post-treatment (SVR12) and analyze the safety profile of these DAAs in a cohort of patients with chronic HCV infection, including those with compensated and decompensated liver cirrhosis. This research aims to support clinical decision-making and improve treatment strategies in real-world settings.

## 2. Methods

### 2.1. Patients

This prospective, multicenter, observational study included patients with chronic hepatitis C virus (HCV) infection who received 12-week SOF/VEL or SOF/VEL/VOX therapy between November 2022 and April 2024 at six high-volume hospitals in South Korea (Appendix A).

SOF/VEL group: Patients were eligible if they (1) had serum HCV RNA > 15 IU/mL, and (2) were aged ≥ 18 years. Exclusion criteria included (1) the presence of residual hepatocellular carcinoma on baseline imaging and (2) the use of concomitant medications contraindicated with SOF/VEL.

SOF/VEL/VOX group: Patients were eligible if they met the following criteria: (1) serum HCV RNA > 15 IU/mL, (2) age ≥18 years, and (3) a history of prior failure with a DAA regimen containing an NS5A inhibitor (genotypes 1–6), or a sofosbuvir-based regimen without an NS5A inhibitor (genotypes 1a or 3).

Patients were excluded if they had (1) hepatocellular carcinoma within six months, (2) contraindicated concomitant medications, or (3) decompensated liver cirrhosis. Decompensation was defined by clinical signs of portal hypertension, such as ascites, total bilirubin > 3 mg/dL, overt hepatic encephalopathy, or variceal bleeding.

### 2.2. Treatment Protocol

Patients in the SOF/VEL group received one tablet of EPUCLUSA^®^ orally per day (400 mg SOF and 100 mg VEL; Gilead Sciences, Foster City, CA, USA) for 12 weeks, while patients in the SOF/VEL/VOX group received one tablet of VOSEVI^®^ orally per day (400 mg SOF, 100 mg VEL, 100 mg VOX; Gilead Sciences, Foster City, CA, USA) for 12 weeks. For patients with decompensated liver cirrhosis treated with SOF/VEL, ribavirin was administered orally twice a day, with food. Dosage was determined by body weight: 400–1000 mg daily for body weight < 75 kg, and 1200 mg daily for body weight ≥ 75 kg. For Child–Pugh B or C patients, the initial dose was 600 mg.

Physical, hematologic, and biochemical examinations were conducted at baseline and then every 4–8 weeks during treatment and follow-up. Adverse events (AEs) were assessed at each clinic visit and were classified according to the Common Terminology Criteria for Adverse Events v.6.0, as presented by the National Cancer Institute Cancer Therapy Evaluation Program [13].

### 2.3. Study Assessments

The primary endpoint of this study was to evaluate the proportion of patients who achieved sustained virological response at 12 weeks post-treatment, defined as undetectable serum HCV RNA using polymerase chain reaction (PCR). HCV RNA levels were measured using a real-time PCR-based method (COBAS TaqMan HCV Test 2.0; Roche Molecular Systems, Pleasanton, CA, USA), with a lower quantification limit of 15 IU/mL across all sites. Virological response was assessed at end of treatment (ETR) and at 12 weeks post-treatment. Treatment failure included nonresponse, breakthrough, relapse, or medication discontinuation.

Nonresponse was defined as detectable serum HCV RNA levels at the evaluation of ETR. Relapse was defined as undetectable serum HCV RNA at ETR that became detectable again during the 12-week post-treatment period.

### 2.4. Evaluation of the Liver Functional Reserve (Liver Fibrosis Score)

In order to evaluate the liver functional reserve before and after SOF/VEL-based treatment, non-invasive liver fibrosis tests were performed. These included the AST-to-platelet ratio index (APRI), the fibrosis-4 (FIB-4) index, and liver stiffness measurement (LSM) using vibration-controlled transient elastography (VCTE, FibroScan^®^, Echosens, Paris, France). These measurements were calculated at baseline and during follow-ups at 4-week intervals until SVR.

### 2.5. Statement of Ethics

This study was approved by the institutional review board of Chonnam National University Hospital (CNUH-2022-41) and those of participating institutions. The study was performed in accordance with the Declaration of Helsinki (2013). All participants provided written informed consent.

### 2.6. Statistical Analysis

SVR rates were assessed in both the full analysis set (intention-to-treat, ITT) and the per-protocol (PP) set, which only included patients who completed the 12-week post-treatment evaluation. Categorical variables were compared using the χ^2^ test or Fisher’s exact test, while continuous variables were analyzed using Student’s *t*-test or the Mann–Whitney U test, as appropriate. Correlations between variables were analyzed using Cox regression analysis, and statistical significance was defined as *p* < 0.05. Analyses were conducted using SPSS version 27.0 (IBM Corp., Armonk, NY, USA).

## 3. Results

### 3.1. Baseline Characteristics of Enrolled Patients

The baseline profiles of the enrolled patients are summarized in Table 1. A total of 101 patients, with a median age of 64.39 and 44.6% being male, underwent SOF/VEL treatment. The most common genotype was 2 (59.4%), followed by 1b (40.6%). Co-infection with HBV and HIV was observed in 6.9% and 2.0% of patients, respectively. The initial mean APRI, FIB-4 index, and LSM were 0.76, 3.18, and 7.45 kPa, respectively. Compensated cirrhosis was present in 21.8% of patients, and decompensated cirrhosis in 5.0%.

Additionally, 17 patients (median age 60.85, 41.2% male) received SOF/VEL/VOX treatment. The most common genotype was 1b (82.4%), followed by 2 (17.6%). HBV co-infection was detected in 5.9% of patients, while the baseline APRI, FIB-4 index, and LSM were 0.86, 2.88, and 7.1 kPa, respectively. Furthermore, 23.5% of patients had compensated cirrhosis, while the most common previously failed DAA regimen was daclatasvir plus asunaprevir (64.7%).

### 3.2. Treatment Efficacy of Sofosbuvir- and Velpatasvir-Based Regimens

The virological response was evaluated at ETR and after 12 weeks of SOF/VEL and SOF/VEL/VOX treatment. In the SOF/VEL group, ETR was achieved in 90.1% of patients, with no available data for the remaining 9.9%. SVR12 was 88.1% in the modified intention-to-treat (mITT) analysis, with all treatment failures due to loss to follow-up (LTFU) (11.9%). In the PP analysis, SVR12 was achieved in 100.0% of patients. In the SOF/VEL/VOX group, ETR was achieved in 94.1% of patients, while the remaining single patient (5.9%) had no available data at ETR. SVR12 was achieved in 94.1% of patients in the mITT analysis with the sole failure due to LTFU. In the PP analysis, SVR12 was achieved in all 17 patients.

### 3.3. Longitudinal Changes in Liver Fibrosis Scores After Treatment

To evaluate the longitudinal changes in liver fibrosis grades before and after DAA treatment, we calculated the APRI, FIB-index, and LSM at baseline, at weeks 4 and 8, and at ETR and SVR12. In the SOF/VEL group, the APRI and FIB-4 improved significantly after four weeks of treatment (1.09 to 0.46, *p* < 0.001, and 4.28 to 2.86, *p* < 0.001, respectively) and demonstrated sustained efficacy throughout SVR12 (Appendix A). Additionally, LSM also showed a significant decrease at SVR12 compared to initial results (10.68 to 9.66 kPa, *p* <0.001) (Figure 1). In the SOF/VEL/VOX group, the APRI and FIB-4 also improved significantly after four weeks of treatment (1.51 to 0.50, *p* < 0.001, and 4.72 to 2.50, *p* < 0.001, respectively) and remained stable through SVR12 (Appendix A). LSM also significantly improved at SVR12 compared to baseline (10.50 to 6.81 kPa, *p* = 0.048) (Figure 2).

### 3.4. Adverse Events During Sofosbuvir- and Velpatasvir-Based Treatment

No grade 3 or 4 AEs were observed in the SOF/VEL group. However, one patient in the SOF/VEL/VOX group experienced grade 3 thrombocytopenia. No treatment-related grade 3–4 AEs or AEs leading to treatment discontinuation occurred in either group. Mild AEs occurred in 12.9% of patients in the SOF/VEL group, including pruritus, headaches, diarrhea, and abdominal discomfort. In the SOF/VEL/VOX group, 35.3% of patients reported mild AEs including cough, epigastric soreness, and headaches. Treatment-related grade 1–2 AEs occurred in 7.9% of SOF/VEL patients (diarrhea, abdominal discomfort, dizziness), and 11.8% of SOF/VEL/VOX patients (headaches, myalgia).

## 4. Discussion

Sofosbuvir- and velpatasvir-based treatments (SOF/VEL and SOF/VEL/VOX) demonstrated excellent efficacy and a favorable safety profile in this prospective, multicenter, observational study. To the best of our knowledge, this is the first prospective real-world study specifically evaluating the effectiveness of SOF/VEL-based therapy for HCV in South Korea. A recent nationwide large-scale retrospective study by Sohn et al. has also demonstrated the impact of DAA-based regimens on the overall HCV disease burden in the Korean population [10]. A prior Phase 3b study conducted in South Korea in 2023, which comprised 53 patients treated with SOF/VEL and 33 patients with SOF/VEL/VOX, reported SVR12 rates of 98.1% and 100.0%, respectively [12].

SOF is a nucleotide-analog NS5B polymerase inhibitor that provides potent antiviral activity and a high genetic barrier to resistance. VEL, an NS5A inhibitor, is pangenotypic and acts synergistically with SOF to suppress viral replication [5,14]. VOX, a protease inhibitor targeting NS3/4A, is added in retreatment settings to overcome resistance, especially in patients with prior failure of NS5A-based regimens [15]. The combination of these agents in fixed-dose formulations ensures simplified dosing and high efficacy across genotypes, including difficult-to-treat populations. In this study, the excellent SVR12 outcomes reaffirm the real-world applicability of these regimens in Korean patients.

In our study, both treatment regimens showed excellent virological outcomes (Table 2). The SVR12 rates were 88.1% in the SOF/VEL group and 94.1% in the SOF/VEL/VOX group based on the mITT analysis, reaching 100% in both groups in the PP analysis. Notably, these favorable outcomes were achieved despite the inclusion of patients with cirrhosis (21.8% with compensated cirrhosis and 5.0% with decompensated cirrhosis in the SOF/VEL group, 23.5% with compensated cirrhosis in the SOF/VEL/VOX group). Additionally, the SOF/VEL/VOX group included patients who had failed various DAA regimens, such as daclatasvir and asunaprevir (64.7%), glecaprevir and pibrentasvir (23.5%), elbasvir and grazoprevir (5.9%), and sofosbuvir and ledipasvir (5.9%), which reflects real-world clinical practice. These findings underscore the potent efficacy of SOF/VEL-based therapies, even in treatment-experienced patients with advanced liver disease.

Our findings are consistent with previous real-world studies from other Asian countries. A multicenter study in Japan reported an SVR12 rate of 100% with SOF/VEL in 37 patients with HCV, including those with compensated cirrhosis [16]. In Taiwan, a nationwide cohort involving 3490 patients also demonstrated a 99.4% SVR12 rate with SOF/VEL, reinforcing its broad efficacy [17]. Additionally, a real-world multicenter study in Taiwan including 107 patients who failed NS5A inhibitor-containing DAA regimens reported an SVR of 100.0% in a PP population. In China, a real-world cohort of 483 patients similarly showed an SVR12 rate of 99.1% in mITT analysis [18]. Another powerful pangenotypic DAA regimen, glecaprevir plus pibrentasvir (GLE/PIB), has also demonstrated excellent treatment efficacy in real-world studies. In a nationwide cohort from Taiwan, an 8-week GLE/PIB regimen achieved an SVR12 rate of 98.2% among treatment-naïve patients with compensated cirrhosis, with favorable tolerability [19]. Similarly, a large multicenter prospective study from Japan involving 1190 patients reported a 99.0% SVR rate in those treated with the 12-week regimen, including patients with cirrhosis, genotype 3, or prior DAA failure [20]. However, while GLE/PIB offers the advantage of a shortened 8-week treatment duration, a critical limitation of GLE/PIB is that it contains a protease inhibitor, making it contraindicated in patients with decompensated cirrhosis. In contrast, SOF/VEL can be safely administered across a broader spectrum of liver disease severity, including decompensated cirrhosis. This therapeutic flexibility reinforces the clinical value of SOF/VEL treatment, particularly in populations with advanced hepatic dysfunction [2,7].

All SVR12 failures in our study were due to LTFU, and no virological relapses were observed. Despite employing a prospective study design, with structured follow-up via regular telephone calls and text message reminders, we were ultimately unable to verify SVR12 in a small subset of patients. This underscores the importance of reliable post-treatment monitoring, as LTFU may lead to an underestimation of treatment failure rates [21,22]. In real-world clinical practice, adherence to SVR12 testing is likely to be even lower than in research settings, suggesting that, although rare, undetected SVR failure may occur [23,24]. Heightened awareness and robust follow-up systems are therefore essential to ensure complete treatment evaluation.

Similar concerns have been raised in previous studies. Dijk et al. highlighted LFTU as a major barrier to achieving global HCV elimination, emphasizing that re-engaging LTFU patients offers a tangible opportunity for micro-elimination strategies, particularly when supported by national frameworks and proactive patient retrieval efforts [25]. Similarly, Darvishian et al. showed that LTFU exceeded viral failure in their real-world study, impeding the cascade of care. Studies report that 0–11% (median 3.4%) of patients become LTFU during therapy and 0–25% (median 4.9%) become LTFU after therapy completion, consistent with our study results [26].

Both SOF/VEL and SOF/VEL/VOX demonstrated favorable safety profiles (Table 3). Treatment-related AEs occurred in 7.9% of SOF/VEL patients and 11.8% of SOF/VEL/VOX patients, indicating a low overall incidence. No treatment-related grade 3–4 AEs occurred in either group. A single case of grade 3 thrombocytopenia in the SOF/VEL/VOX group was deemed unrelated to antiviral therapy and resolved after systemic steroid therapy. Importantly, no AEs led to treatment discontinuation in either group. These findings confirm the overall safety and tolerability of SOF/VEL-based regimens in clinical practice, which is consistent with previous real-world studies [5,6,12,17,27,28].

In the present study, both regimens demonstrated significant improvements in non-invasive liver fibrosis markers. In the SOF/VEL group, APRI and FIB-4 values decreased as early as week 4, which were sustained through SVR12 (Appendix A). Similar trends were demonstrated in the SOF/VEL/VOX group (Appendix A), suggesting that virological clearance contributes to early and sustained regression of liver fibrosis. Notably, LSM, which was assessed by transient elastography, also decreased significantly in both groups, supporting the notion that virological clearance promotes fibrosis regression. These findings are consistent with previous studies, including a study conducted by Krassenburg et al., who reported reductions in liver-related morbidity and all-cause mortality following fibrosis regression in patients with cirrhosis who achieved SVR after DAA therapy [29]. Rockey et al. further emphasized that fibrosis regression after SVR with DAA therapy correlates with improved outcomes, including reduced liver-related complications and HCC risk, making it a meaningful prognostic marker in advanced liver disease.

Taken together, these findings reinforce the notion that DAA-induced virological cure plays a critical role in both histological and clinical recovery in patients with chronic HCV. Beyond a virological cure, successful DAA therapy contributes to liver fibrosis regression, which may ultimately prevent disease progression and reduce HCC risk in treated patients.

This study has several limitations. First, the generalizability of our findings may be limited, as the study was conducted exclusively in South Korea. Second, although efforts were made to minimize LTFU through structured reminders, a proportion of patients did not complete SVR12 assessment, which could potentially lead to an underestimation of failure rates. Third, the study relied on non-invasive markers such as the APRI, FIB-4, and LSM to evaluate fibrosis regression, which, while clinically useful, may not fully capture histopathological changes. Fourth, the statistical power of the subgroup analyses was limited by small sample sizes, particularly among patients with decompensated cirrhosis and those receiving SOF/VEL/VOX after prior DAA failure. Although the number of patients in the SOF/VEL/VOX arm was small (n = 17), this subgroup represents a clinically important treatment failure population, for which real-world data remain limited. Thus, despite its size, this cohort provides valuable insights, consistent with prior Phase 3 studies reporting similarly small samples [30]. Moreover, the overall sample size in our study reflects the treatment landscape in South Korea, where a substantial proportion of patients with HCV had already completed antiviral therapy prior to the availability of SOF/VEL-based regimens. Nevertheless, our study represents the first prospective real-world investigation of SOF/VEL and SOF/VEL/VOX in South Korea, providing meaningful clinical evidence from routine practice settings. However, the small sample size inherently limits the precision of efficacy estimates, as reflected by wide confidence intervals, and precludes stratified analyses by genotype, cirrhosis status, or prior treatment history. These limitations should be considered when interpreting subgroup findings. Future studies with larger sample sizes are warranted to enhance the robustness and generalizability of these findings. All patients treated with SOF/VEL/VOX in our study had previously failed at least one DAA regimen containing an NS5A inhibitor, placing them at high risk of harboring resistance-associated substitutions (RASs) such as L31M/V, Y93H, and P32 deletion [31]. These mutations are known to compromise the efficacy of subsequent treatments, and Itakura et al. demonstrated that multiple treatment failures often lead to complex RAS patterns [32]. Despite this, our SOF/VEL/VOX group achieved an SVR12 rate of 94.1% in the mITT population and 100% in the PP population, indicating that the regimen remains effective even in RAS-presumed patients. Although resistance testing was not performed, future studies incorporating baseline resistance profiling are essential to better understand outcomes in this population. Finally, this study did not include long-term outcomes beyond SVR12, such as hepatocellular carcinoma prevention, liver-related mortality, or the need for liver transplantation, which could preclude the assessment of sustained clinical benefits.

In summary, this prospective, multicenter study demonstrated that SOF/VEL and SOF/VEL/VOX regimens are highly effective, with a favorable safety profile and notable improvements in non-invasive markers of hepatic fibrosis, even among patients with advanced liver disease and prior DAA treatment failure, highlighting their robustness in diverse clinical scenarios. These findings support the adoption of SOF/VEL-based treatments in real-world settings. Nevertheless, long-term prospective studies are warranted to determine whether fibrosis regression translates into sustained clinical benefits, including reduced HCC risk and reduced liver-related morbidity and mortality.

## Figures and Tables

**Figure 1 viruses-17-00949-f001:**
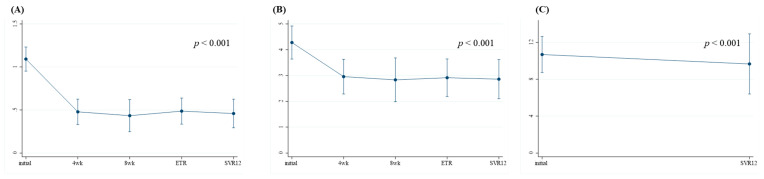
Longitudinal changes in liver fibrosis score—(**A**) APRI score, (**B**) FIB-4 index, and (**C**) Liver stiffness measurement—in patients who were treated with sofosbuvir plus velpatasvir (n = 99).

**Figure 2 viruses-17-00949-f002:**
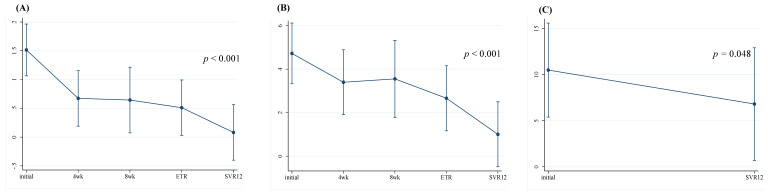
Longitudinal changes in liver fibrosis score—(**A**) APRI score, (**B**) FIB-4 index, and (**C**) Liver stiffness measurement—in patients who were treated with sofosbuvir and velpatasvir plus voxilaprevir (n = 16).

**Table 1 viruses-17-00949-t001:** Baseline characteristics of enrolled patients.

	SOF/VEL (n = 101)	SOF/VEL/VOX (n = 17)
Age (median, range)	64.39 (28.60–86.18)	60.85 (49.00–82.55)
Male (n, %)	45 (44.6%)	7 (41.2%)
Body mass index (kg/m^2^)	23.44 (16.10–33.65)	26.06 (19.85–29.54)
Asian Korea/Ukraine/Uzbekistan/Mongolia	101 (100.0%) 98 (97.0%)/1 (1.0%)/1 (1.0%)/1 (1.0%)	17 (100.0%) 17 (100.0%)/-/-/-
Genotype 1b/2	41 (40.6%)/60 (59.4%)	14 (82.4%)/3 (17.6%)
HTN	40 (39.6%)	7 (41.2%)
DM	23 (22.8%)	3 (17.6%)
Dyslipidemia on statin	15 (14.9%)	3 (17.6%)
Cardiovascular accident	13 (12.9%)	1 (5.9%)
HBV HBsAg/anti-HBc	7 (6.9%)/60 (59.4%)	1 (5.9%)/8 (47.1%)
HIV	2 (2.0%)	-
Platelets (10^3^/μL)	173 (31–526)	167 (35–232)
AST (IU/mL)	45 (14–634)	61 (25–195)
ALT (IU/mL)	39 (5–325)	39 (15–177)
Total bilirubin (mg/dL)	0.61 (0.20–15.56)	0.67 (0.39–1.50)
Albumin (g/dL)	4.20 (2.50–5.10)	4.20 (3.00–4.56)
PT (INR)	1.02 (0.44–3.80)	1.04 (0.95–1.20)
Creatinine (mg/dL)	0.77 (0.42–10.25)	0.73 (0.62–0.94)
APRI	0.76 (0.00–6.14)	0.86 (0.39–5.79)
FIB-4 index	3.18 (0.00–20.83)	2.88 (1.62–17.75)
Liver stiffness (kPa)	7.45 (3.0–50.0)	7.1 (3.7–32.5)
Cirrhosis Compensated/decompensated	22 (21.8%)/5 (5.0%)	4 (23.5%)/-
Previous malignancy history HCC/non-HCC malignancy	6 (5.9%)/9 (8.9%)	1 (5.9%)/1 (5.9%)
Baseline HCV RNA (median, IU/mL)	1,020,000 (745–25,075,260)	1,590,000 (13,732–5,076,892)
Prior interferon treatment	2 (2.0%)	2 (11.8%)
Prior DAA treatment Daclatasvir/asunaprevir Glecaprevir/pibrentasvir Elbasvir/grazoprevir Sofosbuvir/ledipasvir		11 (64.7%) 4 (23.5%) 1 (5.9%) 1 (5.9%)

SOF, sofosbuvir; VEL, velpatasvir; VOX, voxilaprevir; HTN, hypertension; DM, diabetes mellitus; HBV, hepatitis B virus; AST, aspartate transaminase; ALT, alanine aminotransferase; PT, prothrombin time; INR, international normalized ratio; APRI, AST-to-platelet ratio; FIB-4, fibrosis-4; HCC, hepatocellular carcinoma; HCV, hepatitis C virus; DAA, direct-acting antiviral.

**Table 2 viruses-17-00949-t002:** Virological response of sofosbuvir + velpatasvir-based treatment.

	SOF/VEL (n = 101)	SOF/VEL/VOX (n = 17)
ETR HCV not detected Nonresponse No data	91 (90.1%) 0 10 (9.9%)	16 (94.1%) 0 1 (5.9%)
SVR at 12 weeks		
mITT analysis	89/101 (88.1%)	16/17 (94.1%)
Reason for SVR 12 failure Relapse/Virological failure Loss to follow-up	0/0 12 (11.9%)	0/0 1 (5.9%)
PP analysis	89/89 (100.0%)	16/16 (100.0%)

SOF, sofosbuvir; VEL, velpatasvir; VOX, voxilaprevir; ETR, end of treatment response; SVR, sustained virological response; mITT, modified intention-to-treat; PP, per-protocol.

**Table 3 viruses-17-00949-t003:** Adverse events reported during treatment.

	SOF/VEL	SOF/VEL/VOX
All adverse events (AEs)	13 (12.9%) Pruritus (1), headache (1), diarrhea (1), abdominal discomfort (1), dizziness (4), facial rash (1), anemia (2), eczema (1), acute kidney injury (1)	5 (35.3%) Cough (1), epigastric soreness (1), thrombocytopenia (1), headache (1), insomnia (1), myalgia (1)
Grade 3–4 AE	0	1 *
Treatment-related AE	8 (7.9%) Diarrhea (1), abdominal discomfort (1), dizziness (4), anemia (2)	2 (11.8%) Headache (1), myalgia (1)
Treatment-related grade 3–4 AE	0	0
A/E leading to treatment discontinuation	0	0

SOF, sofosbuvir; VEL, velpatasvir; VOX, voxilaprevir. * Grade 3 thrombocytopenia occurred 4 weeks after treatment and resolved after administration of systemic steroid therapy under consultation with hematology.

## Data Availability

The datasets generated during and/or analyzed during the current study are available from the corresponding author upon reasonable request.

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
