# Peer review of "Real-World Treatment Efficacy and Safety Profile of Sofosbuvir- and Velpatasvir-Based HCV Treatment in South Korea: Multicenter Prospective Study"

_viruses, 2025, doi:10.3390/v17070949_

Round 1

Reviewer 1 Report

Comments and Suggestions for Authors

This is an important real-world study evaluating the efficacy of SOF/VEL and SOF/VEL/VOX regimens in Korean HCV patients. However, there are several areas where the manuscript could benefit from further clarification.

1. The small number of SOF/VEL/VOX treated patients

The small number of patients (n=17) in the SOF/VEL/VOX arm is an understandable limitation. However, this subgroup represents a treatment-failure cohort for whom real-world data remain relatively scarce. A prior Phase 3 study (J Viral Hepat. 2021 Sep;28(9):1256–1264. doi:10.1111/jvh.13549) also included a limited number of such cases. In this context, the current Korean real-world data are still valuable and contribute meaningfully to the literature. That said, a larger sample size in future studies would further enhance the robustness and generalizability of these findings.

2. Additional f/u data about decompensated cirrhosis subgroup:

The study includes five patients with decompensated cirrhosis in the SOF/VEL group. It would be helpful if the authors could provide more detailed characterization of this subgroup—such as their specific decompensation features (e.g., ascites, hepatic encephalopathy, variceal bleeding), Child-Pugh class, and treatment outcomes. These details would enhance the clinical interpretability of the data, especially given the heightened risk profile and treatment challenges in this population.

3. Clarification regarding prior hepatocellular carcinoma (HCC):
 The study protocol states that patients with HCC within the past 6 months were excluded. However, the baseline characteristics table indicates that a subset of patients had a history of HCC. It would be helpful if the authors could clarify how these patients were managed prior to enrollment—e.g., whether they underwent surgical resection, local ablation, TACE, or liver transplantation—and whether their HCC was considered cured or stable at the time of antiviral treatment initiation. 

Author Response

Comment 1. The small number of SOF/VEL/VOX treated patients

The small number of patients (n=17) in the SOF/VEL/VOX arm is an understandable limitation. However, this subgroup represents a treatment-failure cohort for whom real-world data remain relatively scarce. A prior Phase 3 study (J Viral Hepat. 2021 Sep;28(9):1256–1264. doi:10.1111/jvh.13549) also included a limited number of such cases. In this context, the current Korean real-world data are still valuable and contribute meaningfully to the literature. That said, a larger sample size in future studies would further enhance the robustness and generalizability of these findings.

Response 1.

We sincerely thank the reviewer for the insightful comment. We fully agree that the small number of patients in the SOF/VEL/VOX group (n=17) is a limitation. However, as the reviewer aptly pointed out, this subgroup represents a treatment-failure population for whom real-world evidence is scarce. In line with previous Phase 3 studies reporting similarly small cohorts, we believe our findings still offer meaningful insights and contribute to the growing body of literature. In response to the reviewer’s suggestion, we have added this point to the limitations section of the revised manuscript and emphasized the need for future studies with larger sample sizes to improve generalizability and statistical robustness.

Comment 2. Additional f/u data about decompensated cirrhosis subgroup:

The study includes five patients with decompensated cirrhosis in the SOF/VEL group. It would be helpful if the authors could provide more detailed characterization of this subgroup—such as their specific decompensation features (e.g., ascites, hepatic encephalopathy, variceal bleeding), Child-Pugh class, and treatment outcomes. These details would enhance the clinical interpretability of the data, especially given the heightened risk profile and treatment challenges in this population.

Response 2.

We thank the reviewer for this thoughtful and important suggestion. In response, we have provided a more detailed characterization of the five patients with decompensated cirrhosis who were treated with SOF/VEL. Specifically, we have included baseline Child-Pugh scores, detailed features of hepatic decompensation (e.g., ascites, hyperbilirubinemia, esophageal varices), and treatment outcomes including post-treatment Child-Pugh scores. These data are now presented in Supplementary Table 3 in the revised manuscript. We agree that such information enhances the clinical interpretability and relevance of our findings, particularly given the complex nature and higher-risk profile of patients with decompensated liver disease.

Comment 3. Clarification regarding prior hepatocellular carcinoma (HCC):
 The study protocol states that patients with HCC within the past 6 months were excluded. However, the baseline characteristics table indicates that a subset of patients had a history of HCC. It would be helpful if the authors could clarify how these patients were managed prior to enrollment—e.g., whether they underwent surgical resection, local ablation, TACE, or liver transplantation—and whether their HCC was considered cured or stable at the time of antiviral treatment initiation.

Response 3.

We thank the reviewer for the thoughtful and important comment. Upon careful review of our study protocol and inclusion criteria, we identified an inconsistency between the stated exclusion criteria in the manuscript and the actual protocol applied. Specifically, patients were excluded if they had residual HCC on imaging (e.g., CT or MRI) at the time of antiviral treatment initiation. We acknowledge this error in phrasing and have revised the manuscript accordingly. We sincerely apologize for any confusion this may have caused.

In accordance with the reviewer’s suggestion, we have now included detailed information regarding patients with a prior history of HCC, all of whom had no radiologic evidence of viable tumor at the time of DAA initiation. These data include the interval from the last HCC-directed therapy to the commencement of antiviral treatment, treatment modalities employed (e.g., TACE, RFA, DEB-TACE), and post-treatment HCC outcomes. This information has been incorporated as Supplementary Table 4 in the revised version of the manuscript.

We are grateful to the reviewer for this valuable feedback, which has allowed us to improve the clarity and scientific rigor of our report.

Reviewer 2 Report

Comments and Suggestions for Authors

Authors treated 17 patients by SOF/VEL/VOX for 12 weeks. The number of patients was too small.

  1. In lines 66-67, “In South Korea, HCV remains a leading cause of chronic liver disease, with genotypes 1b and 2 being the most prevalent.” Please add references. How was HBV??
  2. In lines 72-73,

Make changes from “sustained virological response at 12 weeks post-treatment (SVR12)” to “SVR at 12 weeks post-treatment (SVR12).”

  1. 1. Patients section, authors should use one table, indicating eligible patients and exclude patients.
  2. Authors should rewrite the abstract section and method section in the right way.
  3. In Discussion section, authors should explain SOF in more details.
  4. In Discussion section, authors should explain VOX in more details.
  5. In Discussion section, authors should explain VEL in more details.
  6. Authors should refer the references about SOF/VEL from Asian countries: Japan, China and Taiwan.
  7. Authors should compare the results with those of glecaprevir/piblentasvir from Asian countries.

Author Response

  1. Authors treated 17 patients by SOF/VEL/VOX for 12 weeks. The number of patients was too small.

  We appreciate the reviewer’s comment and agree that the small number of patients treated with SOF/VEL/VOX (n=17) represents a limitation of the study. Nonetheless, this cohort consists of individuals with prior DAA failure, a clinically important population for whom real-world data remain limited. Despite the small sample size, we believe the findings provide valuable preliminary insights that align with existing Phase 3 data, which similarly included only a limited number of such patients. As suggested, we have addressed this limitation explicitly in the revised manuscript and highlighted the need for future studies with larger cohorts to validate and extend these observations.

  1. In lines 66-67, “In South Korea, HCV remains a leading cause of chronic liver disease, with genotypes 1b and 2 being the most prevalent.” Please add references. How was HBV??

  We greatly appreciate your insightful feedback and for highlighting the need to clarify the relative contributions of HBV and HCV to chronic liver disease in South Korea. To address your comment, we have revised the statement on lines 198–199 to reflect that HBV is the primary cause of chronic liver disease, with HCV being the second most common viral etiology. The revised statement now reads “In South Korea, hepatitis B virus (HBV) is the primary cause of chronic liver disease, followed by HCV as a significant viral etiology, with genotypes 1b and 2 predominating” We have also added appropriate references reflecting the etiologies of liver cirrhosis and hepatocellular carcinoma, as well as the distribution of HCV genotypes in the Korean population.

  1. In lines 72-73, make changes from “sustained virological response at 12 weeks post-treatment (SVR12)” to “SVR at 12 weeks post-treatment (SVR12).”

  Thank you for your suggestion. In accordance with your comment, we have revised the phrase in lines 72–73 from “sustained virological response at 12 weeks post-treatment (SVR12)” to “SVR at 12 weeks post-treatment (SVR12)” for clarity and consistency with standard terminology.

  1. Patients section, authors should use one table, indicating eligible patients and exclude patients.

  Thank you for your comment. To enhance clarity, we have added a figure summarizing the patient selection process, including the number of patients screened, excluded, lost to follow-up, and included in the per-protocol analysis. This figure is now presented in the Patients section as suggested (supplementary figure 1).

  1. Authors should rewrite the ab stract section and method section in the right way.

1) Abstract

  Thank you for your valuable comment. In response, we have revised the Abstract to improve clarity regarding patient numbers, treatment grouping, and outcomes. Specifically, we clearly delineated the number of patients treated with SOF/VEL (n=101) and SOF/VEL/VOX (n=17), and specified how many completed follow-up and were included in the per-protocol analysis (89 and 16, respectively). In the Conclusion, we emphasized the prospective nature of the study and the achievement of 100% SVR12 rates, even in patients with cirrhosis or prior treatment failure, to highlight the clinical relevance of our findings. The revised Abstract now better reflects the study’s strengths and implications for real-world clinical practice in South Korea.

Changes have been implemented in the Abstract section of the revised manuscript accordingly.

2) Methods

  Thank you for your valuable suggestion. In response, we have carefully revised the Methods section to enhance clarity, organization, and readability while preserving essential methodological details.

Specifically, we clearly separated the inclusion and exclusion criteria for each treatment group (SOF/VEL vs. SOF/VEL/VOX), improving structural transparency. Definitions of nonresponse and relapse were clarified in the Study assessments subsection. Also typos and unnecessary repetition or technical over-description were revised. We believe that these changes improve the methodological rigor and readability of our manuscript.

The revised Methods section are highlighted on updated manuscript.

  1. In Discussion section, authors should explain SOF in more details.
  2. In Discussion section, authors should explain VOX in more details.
  3. In Discussion section, authors should explain VEL in more details.

Thank you for your insightful suggestion. In response, we have added brief mechanistic and pharmacologic descriptions of sofosbuvir (SOF), velpatasvir (VEL), and voxilaprevir (VOX) in the Discussion section to enhance the reader’s understanding of each agent’s role within the antiviral regimen. These additions emphasize the complementary mechanisms and high efficacy of SOF/VEL and SOF/VEL/VOX, particularly in patients with treatment failure or advanced liver disease. We have also updated the reference list to appropriately cite the pharmacologic characteristics and clinical indications of SOF, VEL, and VOX. The revised text appears in the 2nd paragraph of the Discussion section.

9.Authors should refer the references about SOF/VEL from Asian countries: Japan, China and Taiwan. Authors should compare the results with those of glecaprevir/piblentasvir from Asian countries.

  We thank the reviewer for the insightful suggestion. In response, we revised the Discussion section to include real-world studies on SOFVEL from Japan, Taiwan, and China, showing consistently high SVR12 rates. We also added comparative data on glecaprevir-pibrentasvir from large Asian cohorts. While glecaprevir-pibrentasvir offers the advantage of shorter 8-week treatment, we highlighted its limitation in decompensated cirrhosis. In contrast, SOFVEL can be safely used in a broader range of liver disease, which adds to its clinical value. Relevant references (16–20) have been added accordingly.

Reviewer 3 Report

Comments and Suggestions for Authors

          The authors analyzed the real-world efficacy and safety profile of Sofosbuvir and Velpatasvir against HCV in patients from South Korea. Some concerns should be addressed before acceptance of this manuscript.

  1. The Abstract is not completely clear. The numbers are difficult to understand. For example, there were 101 patients treated with SOF/VEL (it is not clear if also with ribavirin), and from these, 89 were considered in the per-protocol (adherents), excluding 14 (?) non adherent: that sums 103 patients.
  2. Introduction: any reference for lines 68-69?
  3. The authors do not cite a very important paper on the effect of DAAs on HCV disease burden in South Korea (Sohn W et al., Effect of direct-acting antivirals on disease burden of hepatitis C virus infection in South Korea in 2007-2021: a nationwide, multicentre, retrospective cohort study. EClinicalMedicine. 2024 May 30:73:102671. doi: 10.1016/j.eclinm.2024.102671.), with 11725 patients analyzed. This observation is particularly pertinent for the affirmation made on lines 198-199.
  4. In comparison, the number of patients from this study is somehow low, even if the endpoints are not the same.

Author Response

  1. The Abstract is not completely clear. The numbers are difficult to understand. For example, there were 101 patients treated with SOF/VEL (it is not clear if also with ribavirin), and from these, 89 were considered in the per-protocol (adherents), excluding 14 (?) non adherent: that sums 103 patients.

  Thank you for your valuable feedback. We sincerely apologize for the confusion caused by the initial presentation of patient numbers in the Abstract.

We have revised the section to clarify that a total of 101 patients were treated with SOF/VEL. Among them, 12 patients were lost to follow-up, and 89 patients completed the treatment and follow-up, comprising the per-protocol population. SVR12 was achieved in 100.0% (89/89) of these patients.

The revised Abstract now accurately reflects the patient flow and outcomes. We appreciate your thoughtful observation, which helped us improve the clarity of our manuscript.

  1. Introduction: any reference for lines 68-69?

  Thank you for your insightful comment. We fully agree that a reference is needed to support this statement. Unfortunately, there is no publicly available official government documentation or press release specifying the exact date or details of the national insurance coverage or regulatory approval for SOF/VEL and SOF/VEL/VOX in South Korea.

However, we have cited a Phase 3b multicenter Korean study (Heo et al., Korean J Intern Med 2023) that evaluated the efficacy and safety of SOF/VEL and SOF/VEL/VOX regimens in Korean patients. This study was conducted as part of the regulatory approval process and supports the fact that these regimens were implemented in clinical practice in South Korea.

We have now added this reference at the end of the relevant sentence in the Introduction section.

  1. The authors do not cite a very important paper on the effect of DAAs on HCV disease burden in South Korea (Sohn W et al., Effect of direct-acting antivirals on disease burden of hepatitis C virus infection in South Korea in 2007-2021: a nationwide, multicentre, retrospective cohort study. EClinicalMedicine. 2024 May 30:73:102671. doi: 10.1016/j.eclinm.2024.102671.), with 11725 patients analyzed. This observation is particularly pertinent for the affirmation made on lines 198-199.

  Thank you for this helpful comment and for bringing this important study to our attention. We fully agree that the study by Sohn et al. (EClinicalMedicine, 2024), which analyzed 11,725 patients with HCV in South Korea using a nationwide retrospective cohort, provides highly valuable insights into the real-world impact of DAA treatment at the population level.

We have now cited this study in the revised manuscript and clarified that, while Sohn et al. conducted a large-scale retrospective analysis using real-world data, our study represents the first prospective real-world study specifically evaluating the effectiveness of SOF/VEL-based therapy in Korean clinical practice.

We appreciate the reviewer’s suggestion, which has helped us improve the accuracy and context of our statement.

  1. In comparison, the number of patients from this study is somehow low, even if the endpoints are not the same.

  Thank you for your comment. We acknowledge that the number of patients in our study is relatively small compared to international real-world cohorts.

However, this reflects the treatment landscape in South Korea, where SOF/VEL-based regimens were introduced relatively late, and a substantial proportion of HCV patients had already completed antiviral therapy with earlier DAA regimens before these treatments became widely available.

Despite the limited sample size, our study represents the first prospective real-world investigation in South Korea specifically evaluating the effectiveness and safety of SOF/VEL and SOF/VEL/VOX regimens.

We believe that the prospective design and structured follow-up provide meaningful clinical value and complement the existing body of evidence. This point has now been clarified in the limitations section of the Discussion.

Round 2

Reviewer 2 Report

Comments and Suggestions for Authors
  1. " We appreciate the reviewer’s comment and agree that the small number of patients treated with SOF/VEL/VOX (n=17) represents a limitation of the study. Nonetheless, this cohort consists of individuals with prior DAA failure, a clinically important population for whom real-world data remain limited. Despite the small sample size, we believe the findings provide valuable preliminary insights that align with existing Phase 3 data, which similarly included only a limited number of such patients. As suggested, we have addressed this limitation explicitly in the revised manuscript and highlighted the need for future studies with larger cohorts to validate and extend these observations." Authors should describe about this more.
  2. Please expand discussion section more. Several important studies are missing from the references:
    Itakura J, Kurosaki M, Kakizaki S, Amano K, Nakayama N, Inoue J, Endo T, Marusawa H, Hasebe C, Joko K, Wada S, Akahane T, Koushima Y, Ogawa C, Kanto T, Mizokami M, Izumi N. Features of resistance-associated substitutions after failure of multiple direct-acting antiviral regimens for hepatitis C. JHEP Rep. 2020 Jun 18;2(5):100138. doi: 10.1016/j.jhepr.2020.100138. PMID: 32817930
    Nakamoto S, Kanda T, Wu S, Shirasawa H, Yokosuka O.Hepatitis C virus NS5A inhibitors and drug resistance mutations. World J Gastroenterol. 2014 Mar 21;20(11):2902-12. doi: 10.3748/wjg.v20.i11.2902. PMID: 24659881 

Author Response

Reviewer 2

" We appreciate the reviewer’s comment and agree that the small number of patients treated with SOF/VEL/VOX (n=17) represents a limitation of the study. Nonetheless, this cohort consists of individuals with prior DAA failure, a clinically important population for whom real-world data remain limited. Despite the small sample size, we believe the findings provide valuable preliminary insights that align with existing Phase 3 data, which similarly included only a limited number of such patients. As suggested, we have addressed this limitation explicitly in the revised manuscript and highlighted the need for future studies with larger cohorts to validate and extend these observations." Authors should describe about this more.

  We thank the reviewer for this valuable comment. In response, we have revised the manuscript to further elaborate on the implications of the small sample size in the SOF/VEL/VOX subgroup (Discussion, lines 307–321).

 Specifically, we retained the original acknowledgment of the small number of patients in this group and added further detail to clarify the analytical limitations this imposes. We now note that the small sample size inherently limits the precision of efficacy estimates and precludes meaningful stratified analyses by genotype, cirrhosis status, or prior treatment history. These limitations are now explicitly stated to provide a more comprehensive and transparent interpretation of the subgroup findings.

We hope this expanded discussion satisfactorily addresses the reviewer’s concern.

  1. Please expand discussion section more. Several important studies are missing from the references:

Itakura J, Kurosaki M, Kakizaki S, Amano K, Nakayama N, Inoue J, Endo T, Marusawa H, Hasebe C, Joko K, Wada S, Akahane T, Koushima Y, Ogawa C, Kanto T, Mizokami M, Izumi N. Features of resistance-associated substitutions after failure of multiple direct-acting antiviral regimens for hepatitis C. JHEP Rep. 2020 Jun 18;2(5):100138. doi: 10.1016/j.jhepr.2020.100138. PMID: 32817930

Nakamoto S, Kanda T, Wu S, Shirasawa H, Yokosuka O.Hepatitis C virus NS5A inhibitors and drug resistance mutations. World J Gastroenterol. 2014 Mar 21;20(11):2902-12. doi: 10.3748/wjg.v20.i11.2902. PMID: 24659881

  We thank the reviewer for the insightful suggestion regarding resistance-associated considerations in the SOF/VEL/VOX subgroup. In the revised manuscript, we have added a dedicated paragraph to the Discussion section to address this point in greater depth (lines 321-330).

Specifically, we clarified that all patients in the SOF/VEL/VOX group had previously experienced virological failure with at least one NS5A inhibitor–containing DAA regimen (e.g., daclatasvir, pibrentasvir, elbasvir, or ledipasvir), placing them at elevated risk of harboring resistance-associated substitutions (RASs). We referenced key findings from Nakamoto et al. and Itakura et al., which describe the clinical relevance and accumulation of RASs such as L31M/V, Y93H, and P32 deletion after multiple DAA failures

Despite the presumed presence of RASs in our cohort, we observed a high SVR12 rate (94.1% mITT and 100% per protocol), suggesting that SOF/VEL/VOX remains effective even in these treatment-experienced populations. We have also noted the absence of baseline resistance testing in our study and emphasized the need for future prospective investigations incorporating resistance profiling.

We hope this expanded discussion appropriately addresses the reviewer’s recommendation and enhances the scientific rigor and clinical relevance of the manuscript.

Reviewer 3 Report

Comments and Suggestions for Authors

The authors addressed satisfactorely the concerns.

Author Response

The authors addressed satisfactorely the concerns.

Response to Reviewer:
We would like to express our sincere gratitude to the reviewer for the positive feedback and for acknowledging our efforts in addressing the concerns raised. We are grateful for your insightful comments and suggestions, which have contributed to enhancing the clarity and quality of our manuscript.

Round 3

Reviewer 2 Report

Comments and Suggestions for Authors

Authors made corrections accordingly.